# An Optimal WSN Node Coverage Based on Enhanced Archimedes Optimization Algorithm

**DOI:** 10.3390/e24081018

**Published:** 2022-07-23

**Authors:** Thi-Kien Dao, Shu-Chuan Chu, Trong-The Nguyen, Trinh-Dong Nguyen, Vinh-Tiep Nguyen

**Affiliations:** 1Fujian Provincial Key Laboratory of Big Data Mining and Applications, Fujian University of Technology, Fuzhou 350118, China; jvnkien@gmail.com; 2College of Computer Science and Engineering, Shandong University of Science and Technology, Qingdao 266590, China; scchu0803@sdust.edu.cn; 3University of Information Technology, Ho Chi Minh City 700000, Vietnam; dongnt@uit.edu.vn (T.-D.N.); tiepnv@uit.edu.vn (V.-T.N.); 4Vietnam National University, Ho Chi Minh City 700000, Vietnam

**Keywords:** coverage optimization, enhanced Archimedes optimization algorithm, wireless sensor network, optimization approach

## Abstract

Node coverage is one of the crucial metrics for wireless sensor networks’ (WSNs’) quality of service, directly affecting the target monitoring area’s monitoring capacity. Pursuit of the optimal node coverage encounters increasing difficulties because of the limited computational power of individual nodes, the scale of the network, and the operating environment’s complexity and constant change. This paper proposes a solution to the optimal node coverage of unbalanced WSN distribution during random deployment based on an enhanced Archimedes optimization algorithm (EAOA). The best findings for network coverage from several sub-areas are combined using the EAOA. In order to address the shortcomings of the original Archimedes optimization algorithm (AOA) in handling complicated scenarios, we suggest an EAOA based on the AOA by adapting its equations with reverse learning and multidirection techniques. The obtained results from testing the benchmark function and the optimal WSN node coverage of the EAOA are compared with the other algorithms in the literature. The results show that the EAOA algorithm performs effectively, increasing the feasible range and convergence speed.

## 1. Introduction

Wireless sensor networks (WSNs) are mainly composed of several autonomous devices called sensor nodes implemented for specific purposes and scattered in wide areas [1,2]. As wireless communication technology has improved and time has passed [3], WSNs have become more common in the information field [4]. They are utilized in various crucial fields, including the military, intelligent transportation, urban planning, industrial and agricultural automation, and environmental monitoring [5]. The sensor node’s job is to send captured information to the base station (BS) or the destination node by sensing and collecting ambient data, including sound vibration, pressure, temperature, and light intensity, among other things [6]. 

Due to their ease of implementation, cheap maintenance costs, and high flexibility, WSNs have successfully replaced wired networks and been embraced in the industrial field in recent years [7]. However, due to the nature of wireless communication, interference and conflict are invariably present during data transmission [8], and data packets may be lost or delayed past their planned deadline [9].

One of the most fundamental difficulties in WSNs is coverage, which is a critical metric for evaluating coverage optimization efforts. Because coverage affects the monitoring capability of the target monitoring area, it substantially impacts WSNs’ quality of service [10]. A node coverage optimization technique has been developed to increase the coverage of wireless sensor nodes in a big-data environment, considering the characteristics of large wireless sensor networks with limited node computing capabilities [11,12]. However, wireless sensor networks’ operating environment is complex and changing, and sensor energy is limited and cannot be supplemented [13]. 

Deployment of sensor nodes that is both sensible and effective reduces network expenses and energy consumption [14]. All WSN coverage applications try to deploy a minimal number of sensor nodes to monitor a defined target region of interest to improve coverage efficiency. Sensor nodes are typically placed randomly in the target monitoring region, resulting in an uneven distribution of nodes and limited coverage [15]. As a result, the network coverage control problem is the central research problem in wireless sensor networks [16]. Adopting an effective and acceptable network coverage control technique is beneficial to optimizing sensor node deployment to increase wireless sensor network performance. Sensor nodes are randomly placed around the monitoring region [17]. Strategically positioning sensor nodes in the monitoring zone is crucial to increasing WSN node coverage. For large-scale sensor node deployment challenges, logical and efficient deployment of WSNs has been demonstrated to be an NP-hard problem, and finding the best solution remains challenging [18]. Multiple nodes must be deployed to meet the monitoring needs, resulting in significant network redundancy coverage issues, the repeated transmission of vast amounts of data in the network, and a rise in the number of network nodes [19].

The metaheuristic algorithm is one of the promising approaches being examined as a solution for dealing with WSN node coverage in this scenario [20]. Metaheuristic algorithms can identify near-optimal solutions in a fair amount of time with limited nodes and computational resources, making them a convenient approach to the WSN coverage optimization problem [21]. Approximation optimization techniques with solutions that can tackle high-dimensional optimization problems effectively are known as metaheuristic algorithms [22]. Natural phenomena, such as human behaviors, physical sensations, animal swarm behaviors, and evolutionary concepts, are frequently used to inspire metaheuristic algorithms [23]. The metaheuristic optimization algorithms are widely used in a variety of fields, including technology, health, society, and finance, and are especially good at meeting time deadlines [20]. They are usually fairly easy to implement, having few parameters, being relatively simple to understand, and powerful, including selecting for biological nature, natural social swarm behavior, and autocatalytic physical phenomena, e.g., simulated annealing (SA) [24], genetic algorithms (GAs) [25,26], particle swarm optimization (PSO) [27], cat swarm optimization (CSO) [28], parallel PSO (PPSO) [29], ant colony optimization (ACO) [30], artificial bee colony (ABC) [31], bat algorithms (BA) [32,33], moth–flame optimization (MFO) [34,35], whale optimization algorithm (WOA) [36], flower pollination algorithm (FPA) [37,38], sine–cosine algorithm (SCA) [39,40], etc.

A new metaheuristic optimization method based on suggested physical laws is the Archimedes optimization algorithm (AOA) [41], which is mimicked by the location update technique that uses object collisions for processing optimization equations. The optimization is carried out by modeling Archimedes’ buoyancy principle process: following a crash, the object progressively assumes neutral buoyancy. The AOA has advantages and the potential to optimize various engineering problems because of its fewer parameters, making it more easily understandable in programming. However, there are specific problems with the AOA algorithm approach to particular issues, such as the solution’s slow convergence time and poor quality. 

This paper suggests an enhanced Archimedean algorithm (EAOA) for the global optimization problems and node coverage optimization in WSN deployment. The difficulties of WSN nodes’ uneven distribution and low coverage in the random deployment of WSN monitoring applications are approached based on the EAOA. The entire WSN monitoring area can be divided into multiple sub-areas, and then node optimization coverage can be implemented in each sub-area based on evaluating the objective function values. The modeled objective function is calculated by the all-nodes coverage area ratio of the probability of the deployed surface 2D WSN monitoring area of the network. We implemented the EAOA by adapting its updating equations using reverse learning and multidirection strategies to overcome the limitations of its original approach. The following item list briefly highlights the contributions of this paper’s innovations:Offering strategies for enhancing the AOA to prevent the original algorithm’s drawbacks in dealing with complex situations, evaluating the recommended method’s performance by using the CEC2017 test suite, and comparing the proposed method’s results with the other algorithms in the literature.Establishing the objective function of the optimal WSN node coverage issues in applying the EAOA and AOA for the first time, and analyzing and discussing the results of the experiment in comparison with swarm intelligence optimization algorithms.

The paper’s remaining parts are organized as follows: Section 2 describes the WSN node coverage model as a statement problem, and reviews the AOA algorithm as related work. Section 3 presents the proposed EAOA, and evaluates its performance under the test suite. Section 4 offers the EAOA for tackling the node coverage issues by applying the EAOA algorithm and analyzing the simulation results. The conclusions are presented in Section 5.

## 2. System Definition

This section presents the WSN node coverage model as the problem statement, and the original algorithm—called the Archimedes optimization algorithm (AOA)—as a recent metaheuristic optimization algorithm. The subsections are reviewed as follows.

### 2.1. WSN Node Coverage Model

The coverage optimization problem is the desired location of each deployed node, with a fixed sensing radius for each sensor. Each node needs to be deployed with a limited sensing radius, and each sensor can only sense and find within its sensing radius. Detection within its sensing radius is a workable solution to the coverage problem. Assuming that the WSN is deployed in a two-dimensional (2D) monitoring area of *W × L* m^2^, with M nodes set up randomly [15,42], then if *S* is a set of nodes denoted as $S=\left\{S_{1},\;S_{2},\;\ldots ,\;S_{i},\;\ldots ,\;S_{M},\;i=1,\;2,\;\ldots ,\;\mathrm{and}\;M\right\}$, the coordinates of each node $S_{i}$ can be represented as ($x_{i},\;y_{i}$). A sensor node’s sensing range is a circle, with the center of the sensing radius $R_{s}$ as its radius. The model of a two-dimensional WSN monitoring area network is assumed as follows:
The sensing radius of each sensor node is $R_{s}$, and the communication radius is $R_{c}$, both measured in meters, with $R_{c}\geq 2R_{s}$. The sensor nodes can normally communicate, have sufficient energy, and can access time and data information.The sensor nodes have the same parameters, structure, and communication capabilities.The sensor nodes can move freely and update their location information in time.


Let T be a set of target monitoring points, $T=\{T_{1},\;T_{2},\;\ldots ,\;T_{j},\;\ldots ,\;T_{n}\},\;j=1,\;2,\;\ldots ,\;n$; the $T_{j}$ coordinate is ($x_{j},\;y_{j}$) in the two-dimensional WSN monitoring area. If the distance between the target monitoring point $T_{j}$ and any sensor node is less than or equal to the sensing radius $R_{s}$, then $T_{j}$ is covered by the sensor nodes. With the sensor node $S_{i}$ and goal monitoring point $T_{j}$, the Euclidean distance is defined as follows:(1)$$d\left(S_{i}{,\;T}_{j}\right)=\sqrt{{{(x}_{i}-x_{j})}^{2}{\;+\;(y}_{i}-y_{j}{)}^{2}},$$
where $d\left(S_{i},T_{j}\right)$ is the distance from node $S_{i}$($x_{i},\;y_{i})$ to node $T_{j}$($x_{j},\;y_{j})$. The node sensing model is set on the sensing radius if $R_{s}$ is greater than or equal to $d\left(S_{i},T_{j}\right)$—the probability p that the target is set to 1; otherwise, it is set to 0. The probability formula is given as follows:(2)$$p\left(S_{i},T_{j}\right)=\{\begin{array}{c} 1,\;\;\;\;\;\;\;\;\;\;\;R_{s}\;\geq d\left(S_{i},T_{j}\right)\\ 0,\;\;\;\;\;\;\;\;\;\;\;R_{s}<d\left(S_{i},T_{j}\right) \end{array}\;\;\;,\;\;$$
where $p\left(S_{i},T_{j}\right)\;$is the probability between the sensor node $S_{i}$ and goal monitoring point $T_{j}$. The sensor nodes can work cooperatively by affecting the neighbor nodes of the deployed two-dimensional WSN monitoring area. Whenever any target monitoring point can be covered by more than one sensor simultaneously, the probability of monitoring the target point $T_{j}$ is given by the following formula:(3)$$P\left(S,\;T_{j}\right)=1-\overset{M}{\underset{i=1}{\prod }}\left(1-p\left(S_{i},T_{j}\right)\right),$$
The ratio of the total area covered by all sensor nodes in the monitoring area to that area’s overall size is known as the coverage rate. Accordingly, the probability ratio to the network’s surface 2D WSN monitoring area is used to calculate the coverage ratio.
(4)$$Cov_{R}=\frac{\sum_{j=1}^{M}P\left(S,T_{j}\right)}{W\times L},$$
where $Cov_{R}$ is the WSN nodes’ coverage ratio in the target point reaching area, $P\left(S,T_{j}\right)$ is the probability of the target point reaching sensed node monitoring, and W×L is the deployed area of the desired surface 2D network.

### 2.2. Archimedes Optimization Algorithm (AOA)

The AOA is a recent metaheuristic optimization algorithm based on Archimedes’ buoyancy principle’s physical principles [41]. The position of its object is updated by imitating the process of the object gradually exhibiting neutral buoyancy following a collision. The AOA algorithm provides the individual population by immersing objects with volume, density, and acceleration properties. The items can determine their position in the fluid based on these attributes. The characteristics and places of the object are randomly initialized at the start of the process. The AOA updates the object’s volume, density, and acceleration during processing optimization. The object’s position is updated based on its individual qualities. Initialization, updating object properties, updating the object’s status, and evaluation are the significant processing steps of the AOA.

***Initialization of the position and attributes*** of the object is conducted as follows: (5)$$X_{i}{=lb}_{i}+rand()\cdot \left({ub}_{i}-{lb}_{i}\right),$$
where $X_{i}$ is a candidate solution vector i-th of the object population size N, i=1, 2, …, N; the boundaries $lb_{i}$ and $ub_{i}$ are the upper and lower boundaries, respectively; and the variable rand() is a d-dimensional vector generated randomly between [0, 1]. The variables of acceleration, volume, and density of the i-th object are noted as $ac_{i}$, $vo_{i}$, and $de_{i}$, respectively; ${\mathrm{vo}}_{i}=\mathrm{rand}()$, ${\mathrm{de}}_{i}=\mathrm{rand}()$, and $acc_{i}=lb_{i}+\mathrm{rand}()\cdot \left({\mathrm{ub}}_{i}-{\;\mathrm{lb}}_{i}\right)$. The position and attributes of the optimal object—such as $X_{best}$, $de_{best}$, $vo_{best}$, and $ac_{best}$—are the selected objects with the best fitness values according to the evaluation of each object.

***Updating object properties*** phase: During the iteration, the volume and density of the object are updated according to the following formula:(6)$${\mathrm{vo}}_{i}^{t+1\;}{=\mathrm{vo}}_{i}^{t}+\mathrm{rand}\cdot \left({\mathrm{vo}}_{\mathrm{best}}-{\;\mathrm{vo}}_{i}^{t}\right),$$
(7)$${\mathrm{de}}_{i}^{t+1}{=\mathrm{de}}_{i}^{t}+\mathrm{rand}\cdot \left({\mathrm{de}}_{\mathrm{best}}-{\mathrm{de}}_{i}^{t}\right),$$
where $vo_{i}^{t+1}$ and $de_{i}^{t+1}$ denote the volume and density of the i-th object in the t+1 iteration, respectively. The simulated collisions between objects in the AOA are mimicked for the optimization process; as time goes on with iterations, the algorithm gradually reaches equilibrium. A transform variable is used as a simulation of the process to realize the algorithm’s transformation from searching exploration to exploitation, as follows:(8)$$\mathrm{TF}=\exp\left(\frac{t-t_{\max}}{t_{\max}}\right)$$
where TF is the transition transform variable, while $t_{max}$ and t are the maximum number of iterations and the current number of iterations, respectively. TF gradually increases to 1 over time. TF≤0.5, meaning that one second of the iteration is in the exploration phase. The update acceleration of object attributes is related to the collision objects.
(9)$${\mathrm{ac}}_{i}^{t+1}=\{\begin{array}{c} \frac{{\mathrm{de}}_{\mathrm{mr}}{\;+\;\mathrm{vo}}_{\mathrm{mr}}\cdot {\mathrm{ac}}_{\mathrm{mr}}}{{\mathrm{de}}_{i}^{t+1}\;\cdot {\;\mathrm{vo}}_{i}^{t+1}},\;\;\mathrm{if}\;TF\leq 1/2\\ \frac{{\mathrm{de}}_{\mathrm{best}}{\;+\;\mathrm{vol}}_{\mathrm{best}}\cdot {\mathrm{ac}}_{\mathrm{best}}}{{\mathrm{de}}_{i}^{t+1}\;\cdot {\;\mathrm{vo}}_{i}^{t+1}},\;\mathrm{otherwise}\; \end{array}$$
where $de_{mr}$, $vo_{mr}$, and $ac_{mr}$ are the density, volume, and acceleration of random material (mr), respectively. If TF≤0.5, there is a collision between objects, and the acceleration updates the formula of object i in iteration t; otherwise, there is no collision between objects. The normalization strategy for the acceleration can be updated as follows:(10)$${\mathrm{ac}}_{i,\mathrm{norm}}^{t+1}=\mathrm{ur}\cdot \frac{{\mathrm{ac}}_{i}^{t+1}-\min\left(\mathrm{ac}\right)}{\max\left(\mathrm{ac}\right)\;-\;\min\left(\mathrm{ac}\right)}+\mathrm{lr},\;$$
where $ac_{i,norm}^{t+1}$ represents the normalized acceleration of the *i*-th object in the t+1 iteration, while ur and lr are the normalized ranges, which are set to 0.8 and 0.2, respectively. 

***Updating the objects’ position*** is conducted as follows: If TF≤1/2 (exploration phase), the position update formula of object *i* at the t+1 iteration is helpful to search from global to local and converge in the region where the optimal solution exists; otherwise, it is a searching exploitation phase for the positional updating. When the object is far from the best position, the acceleration value is enormous, and the object is in the exploration phase. When the acceleration value is small, the object is close to the optimal solution. The exploitation phase can be described as follows:(11)$$X_{i}^{t+1}{\;=X}_{i}^{t}{\;+\;C}_{1}\cdot \mathrm{rand}\cdot {\mathrm{ac}}_{i,\mathrm{norm}}^{t+1}\cdot d\cdot \left(X_{\mathrm{rand}}-X_{i}^{t}\right)$$
where $C_{1}$ is a constant that is set to 2, and d is the density factor that decreases over time, i.e., $d=\exp\left(\frac{t\;-\;t_{\max}}{t_{\max}}\right)-\left(\frac{t}{t_{\max}}\right)$. The acceleration changes from big to small, indicating the algorithm’s transition from exploration to exploitation, respectively, which helps the object approach the optimal global solution.
(12)$$X_{i}^{t+1}{\;=X}_{\mathrm{best}}^{t}+F\cdot C_{2}\cdot \mathrm{rand}\cdot {\mathrm{ac}}_{i,\mathrm{norm}}^{t+1}\cdot d\cdot \left(T\cdot X_{\mathrm{best}}-X_{i}^{t}\right)$$
where $C_{2}$ represents the constant *t*; T is a variable proportional to the transfer operator—the percentage used to attain the best position—$T=C_{3}\times TF$; and F is the direction of motion, and its expression is as follows:(13)$$F=\{\begin{array}{c} +1,\;\mathrm{if}\;P\;\leq \;0.5\\ -1,\;\mathrm{if}\;P\;>\;0.5 \end{array}$$
where $P\;\mathrm{is}\;\mathrm{set}\;\mathrm{to}\;2\cdot rand-C_{4}$. 

***Evaluating*** objective function involves computing the fitness values for the objective function after updating the object’s position each iteration time. The model with objective function is used for fitness value evaluation by evaluating each object that is recorded with the best fitness value found in each position, e.g., $X_{best}$, $de_{best}$, $vo_{best}$, and $ac_{best}$ are updated for the next iterations or generations.

## 3. Enhanced Archimedes Optimization Algorithm

In order to enhance the population of diverse objects, an enhanced version of the Archimedes optimization algorithm (EAOA) based on the opposing learning and diversity guiding techniques is presented in this section. The suggested processes are offered first, followed by a detailed presentation of the evaluation and discussion findings.

### 3.1. Enhanced Archimedes Optimization Algorithm

The AOA is a new metaheuristic algorithm with several advantages, including ease of understanding and implementation, along with local search capability. Still, it has drawbacks, such as jumping out of the optimal local operation, slow convergence, or vulnerability to local optima when dealing with complex problems, such as optimal WSN node coverage issues.

***A multiverse-directing strategy:*** In the original expression in Equation (13), the direction of motion *F* has just two motion directions. For complicated problems, the space may have more scales in terms of motion in space. We can exploit this to increase the number of search directions in complex spaces. A variable of the direction guiding factor G is used as an equivalent to the direction value. An alternative formula of motion direction can be expressed as follows:(14)$$F_{\mathrm{new}}=\{\begin{array}{l} +G\cdot \mathrm{rand}(),\;\mathrm{if}\;P\;\leq \;0.5\\ -\mathrm{tG}\cdot \mathrm{rand}(),\;\mathrm{otherwise} \end{array},$$
where $F_{new}$ is an alternative direction guiding factor, and rand() is a random number ∈ [0, 1] for making the different search values of directions.

***Opposite direction strategy:*** The original and reversed solutions are sorted fitness values based on objective function issues to convert objects in a seeking, forward-exploiting procedure in the optimization problem space. The agents in the optimization space can swiftly converge to the task of the ideal solution by identifying new objects with the best fitness ratings by using direct vetting or other optimization strategies to establish new things in the solution space. A new solution set can be generated by applying reverse learning with a specific rate to join the original for further optimization.

Let $S\left(x_{1},\;x_{2},\;\ldots ,\;x_{i},\;\ldots ,\;x_{D}\right),$ and $S^{\prime }\left(x_{1}^{\prime },\;x_{2}^{\prime },\;\ldots ,\;x_{D}^{\prime }\right)$ be solutions of forwarding and corresponding inverse sets, where$\;x_{i}\in \left[a_{i},b_{i}\right]$, (i=1, 2, …, d). A range [a,b] of the opposite solution set can be expressed as $x_{i}^{\prime }=a_{i}+b_{i}-x_{i}$. The same idea of the opposite learning applied to a new solution is as follows: (15)$$S_{inew}^{\prime }=S\cdot \beta_{r},$$
where $\beta_{r}$ is a variable as an adjustment coefficient for generating and affecting a new solution object set. A portion of the worst solution—e.g., about 15% of sorting values of evaluation object positions—is eliminated to be used for generating a new object set in dimension *d* of the solution space. The adjustment coefficient is calculated as follows:(16)$$\beta_{r}{=R}_{\mathrm{istar}}\cdot \frac{\mathrm{rand}\left(\beta ,\gamma \right)}{D},$$
where rand() is a random function in the range from β to γ. In the experiment, β can be set to −0.5 and γ set to 0.5. *D* is the dimension of problem space, while $R_{istar}$ is the distance between the ideal solution and the one that is closest to optimal. The adjustment coefficient can be applied to the exploiting search of the algorithm for generating and affecting a new solution object set merged into Equation (17).

The strategies and equations of reverse learning $\beta_{r}$ and multiverse-directing $F_{new}$ can be hybridized into updated formulas for generating new solutions. An update of the position of the objects is conducted as follows:(17)$$x_{i}^{t+1}=\{\begin{array}{ll} x_{i}^{t}+\beta_{r}\cdot C_{1}\cdot rand\cdot ac_{i,norm}^{t+1}\cdot d\cdot (x_{rand}-x_{i}^{t}), & if\;TF\;\leq \;0.5\\ x_{\mathrm{best}}^{t}\;+\;F_{\mathrm{new}}\cdot C_{2}\cdot \mathrm{rand}\cdot ac_{i,norm}^{t+1}\cdot d\cdot (T\;\times \;x_{best}-x_{i}^{t}), & \mathrm{otherwise} \end{array}$$

Algorithm 1 depicts the pseudocode of the enhanced Archimedes optimization algorithm (EAOA).
**Algorithm 1** A pseudocode of the EAOA.1. **Input:**$N_{P}$: The population size, *D*: dimensions, *T*: the Max_iter, *C*_1_, $C_{2}$, $C_{3}$,$\;C_{4}$: variables, and *ub*, *lb*: upper and lower boundaries.2. **Output:** The global best optimal solution.3. **Initialization:** Initializing the locations, *vol.*, *de.*, and *acc.* of each object in the population of Equation (8); obtaining each object’s position by calculating the objective function, and the best object in the population is selected; the iteration *t* is set to 1.4. **While**t<T **do**5.   **For** $i=1:N_{p}$ **do**6.     Updating *vol.*, and *de.*, of the object by Equations (6) and (7).7.     Updating TF- transfer impactor and d-*de*., variables are by Equation (8).       **If** TF≤1/2 **then**8.         Updating *acc.* the object acceleration by Equation (10).9.         Updating the local solution by Equation (11).10.       **Else**11. Updating the object accelerations by Equations (9) and (10).12. Updating global solution position by Equation (17).13. **End-if**14.     **End-for**15. **End-while**16. Evaluating each object with the positions and17.   Selecting the best object of the whole population.18. Recording the best global outcome of the optimal object.19. *t*-iteration is set to *t* + 120. **Output:** The best object optimization of the whole population size.

### 3.2. Experimental Results for Global Optimization

The suggested algorithm’s potential performance needs to be tested and verified with the benchmark functions. The CEC 2017 [43] test suite has 29 different test functions to evaluate the EAOA algorithm. There are various types of complexity and dimension functions in the test suite, e.g., f1~f3: unimodal, f4~f10: multimodal, f11~f20: hybrid, and f21~f29: compound test functions. The achieved results of the EAOA are compared not only with the original AOA [41], but also with other selected popular algorithms in the literature, e.g., genetic algorithms (GAs) [25], simulated annealing (SA) [24], particle swarm optimization (PSO) [27], moth–flame optimization (MFO) [34], improved MFO (IMFO) [35], flower pollination algorithm (FPA) [37], sine–cosine algorithm (SCA) [39], enhanced SCA (ESCA) [40], parallel PSO (PPSO) [29], and parallel bat algorithm (PBA) [33]. An expression of $\Delta f=f_{i}-f_{i}^{*}$ is a different error value between the function minimum value $f_{i}^{*}$ and the obtained result value $f_{i}$ of the *i*-th function. The fundamental conditions are set for all algorithms to ensure that the experiment is fair, e.g., the population size is set to 40; the maximum number of iterations is set to 1000; the number of dimensions is set to 30; the solution range of all of the test functions is set to [−100, 100]; and the number of runs is set to 25. Table 1 lists the basic parameters of each algorithm.

The obtained outcomes of the proposed EAOA approach can be verified by several cases—such as the affected strategies with the original algorithm—and compared with the other algorithms. First, the outcomes of several implemented tactics are contrasted with those of the original AOA algorithm. The findings from the EAOA are then contrasted with those from other algorithms. Table A1 compares the affected strategies in applying the EAOA with the original AOA algorithm, and verifies the impact of the suggested techniques used in the EAOA compared to the original AOA algorithm. The data values of the mean outcomes of 25 runs show the best obtained global optimal results, as well as the data on runtime and CPU execution. It can be seen that in some cases strategies 1 and 2 are better than the original algorithm. In most test function cases, the combined strategies 1 and 2 in the proposed EAOA can produce better results than the AOA, and the runtime is not much longer than that of the AOA.

Moreover, the obtained results from the EAOA were also further evaluated to verify the proposed approach’s performance in the presentation. The findings of the EAOA compared with the other algorithms—e.g., GA [25], PSO [27], BA [32], PPSO [29], MFO [34], and WOA [36] algorithms—are presented in Table A2, Table A3, Table A4 and Table A5 and Figure A1. The data values in Table A2, Table A3 and Table A4 are the Mean, Best, and Std.—a standard deviation that measures variables for analyzing the algorithm’s performance. The values of Mean, Best, and Std. are for assessing the search capability, quality, and resilience of the algorithm, respectively. Table A2, Table A3 and Table A4 compare the results of the proposed EAOA with the other popular metaheuristic algorithms in the literature, e.g., the GA [25], PSO [27], BA [32], PPSO [29], MFO, [34], and WOA [36] algorithms. The highlighted data values in each row of Table A2, Table A3 and Table A4 are the best in each pair comparing the EAOA-obtained results with the others for testing functions with a suitable format and layout. The symbols Win, Loss, and Draw at the end of each table provide a brief statistical summary. It can be seen that the proposed EAOA algorithm has the highest number of ‘Wins’. This means that the EAOA produces better results than the other algorithms, and that the EAOA has an excellent optimization performance. 

Figure A1 compares the convergence outcome curves of the EAOA with the ESCA [40], IMFO [35], AOA [41], PPSO [29], WOA [36], and PBA [33] algorithms for the selected functions. The Y coordinate axis represents the average of 25 runs of the best output of the algorithms thus far. The X coordinate shows the iteration in the generation of searching methods. It can be seen from the figure that the EAOA performance curve shows a faster convergence rate than the other algorithms.

Furthermore, for another view of the evaluation results of the proposed approach, we applied the Wilcoxon signed-rank technique for ranking the outcomes. This test compares the pairwise algorithms’ results between the EAOA and other enhanced methods—e.g., PBA, WOA, PPSO, AOA, IFMO, and ESCA algorithms—under the Wilcoxon signed-rank test. Table A5 lists the results of comparison of the pairwise algorithms’ results between the EAOA and other algorithms when applying the Wilcoxon signed-rank test. The bold-highlighted results in Table A5 are the outcomes with p < 0.05. It can be seen that most values have p < 0.05, indicating that the optimization results of the EAOA are significantly different from those of the other algorithms. The average ranking value is 2.25204, and the lowest output of the EAOA is superior to that of the other algorithms. In general, it can be seen that the proposed EAOA can compete with some of the other popular algorithms.

## 4. Optimal WSN Node Coverage Based on EAOA

This section demonstrates how the EAOA algorithm can be used to deploy a WSN with the best node coverage possible, followed by a subsection covering the majority of the processing stages, analysis, and discussion of the findings.

### 4.1. Optimal Node Coverage Strategy

The feasible solution to the optimal node coverage problem is the deployment of each node with a limited sensing radius, where each sensor can only sense and find within its sensing radius. The finding within its sensing radius is a workable solution to the coverage optimization problem. Assuming that the sensing radius of all nodes is the same, and the sensing radius of the node r≤R, any point in the monitoring area is covered if it is located within the sensing radius of at least one sensor node. The monitoring area is divided into the coverage area and the blind spot. Any point in the coverage area is covered by at least one sensor node, and the blind spot complements the coverage area. Some applications need to monitor events with high accuracy. Any point in the coverage area must be at least within the sensing radius of M nodes simultaneously; otherwise, it will be regarded as a blind spot, which we call M double-coverage. The location-seeking process of nodes is abstracted as the process of implementing varied movement behaviors of the object group toward food or a specific site. 

The purpose of WSN coverage optimization utilizing the EAOA approach is to optimize the coverage of the target monitoring area by using a limited number of sensor nodes and optimizing their deployment locations. Let F(x) be the objective function of the WSN nodes’ coverage optimization; the coverage ratio, which is the maximum ratio of probability to the network’s deployed surface 2D WSN monitoring area, is used to determine the objective function for the optimization problem. The maxima are as follows according to Equation (4):(18)$$F\left(x\right)=Maximize\;Cov_{R}=\frac{\sum_{j=1}^{M}P\left(S,T_{j}\right)}{W\times L},$$
where $Cov_{R}$ and $P\left(S,T_{j}\right)$ are the coverage ratio of the WSN nodes and the probability of the target point reaching W×L in the sensed node of the 2D monitoring network’s deployed area, respectively. Each individual object in the algorithm represents a coverage distribution, and the specific steps of the algorithm scheme for the coverage optimization are listed as follows:
Step 1: Input parameters such as a number of nodes M, perception radius $R_{s}$, area of region W×L, etc.Step 2: Set the parameters of population size *N*, the maximum number of iterations *max_Iter*, the density factor, and prey attraction, and randomly initialize the object’s positions using Equations (5)–(7).Step 3: Enhance the initializing population—the parameters of Equations (8)–(10), (14), and (15)—and calculate the objective function for initial coverage according to Equation (18).Step 4: Update the position of objects and the strategy according to Equation (17), and then compare them to select the best fitness value according to the objective function value.Step 5: Calculate the individual values of objects and retain the optimal solution of the global best.Step 6: Determine whether the end condition is reached; if yes, proceed to the next step; otherwise, return to Step 4.Step 7: The program ends and outputs the optimal fitness value and the object’s best location, representing the node’s optimal coverage rate outputs.


### 4.2. Analysis and Discussion of Results

The scenarios assuming that the WSN’s sensor nodes are deployed in a square monitoring area of *W × L* can be set to scenario areas, e.g., 40 m × 40 m, 80 m × 80 m, 100 m × 100 m, and 160 m × 160 m. Table 2 lists the experimental parameters of the WSN node deployment areas. The sensing radius of sensor nodes $R_{s}$ is set to 10 m. The communication radius $R_{c}$ is set to 20 m. The number of sensor nodes is denoted by *M*, consisting of 20, 40, 50, and 60 sensor nodes. *Iter* indicates the number of iterations, which may be set to 500, 1000, or 1500.

The optimal results of the EAOA were compared with the other selected schemes—i.e., the SSA [44], PSO [45], GWO [46], SCA [47], and AOA [48]—for the coverage optimization of WSN node deployment to verify the adequate performance of the algorithm. Figure 1 displays a graphical diagram of the nodes’ initialization with the EAOA for the statistical coverage optimization scheme with different numbers of sensor nodes: (a) 20, (b) 40, (c) 50, and (d) 60.

Table 3 compares the proposed EAOA approach to other strategies—i.e., the SSA, PSO, GWO, SCA, and AOA algorithms—in terms of percentage coverage rate, running time, convergence iterations, and monitoring area size. It can be seen that the EAOA scheme produces the best global solution in the coverage areas, with a high coverage rate, coverage of the node’s whole area, and a faster runtime than the other approaches.

Figure 2 indicates the graphical coverage of six different metaheuristic algorithms—i.e., the AOA, SSA, PSO, GWO, SCA, and EAOA approaches—for the WSN node area deployment scenarios for optimal coverage rates, with the same density and environmental setting conditions. Because the EAOA algorithm can avoid premature phenomena, its coverage rate is reasonably high, with less overlap. It can better alter the node configuration than the other competitors for the monitoring area’s network coverage. The graphics show the differences in the distribution of coverage; the differences are so small that the graphics look very similar, with the graph of node coverage distribution showing seemingly identical results. Furthermore, Figure 3 and Figure 4 show that the convergence curves of the proposed EAOA approach can provide higher percentages of statistical coverage than the other methods used.

Figure 3 indicates four different sizes of WSN monitoring node area deployment scenarios of the metaheuristic approaches for optimal coverage rates. The convergence curves of the proposed EAOA approach can provide higher percentages of statistical coverage than the other methods used. 

Figure 4 shows the coverage rate of the EAOA compared against the SSA, PSO, GWO, SCA, and AOA algorithms for statistical sensor node count deployment for the 2D monitoring of different areas. It can be seen the EAOA algorithm produces a coverage rate that is reasonably high in the monitoring area’s network coverage. The results show that the EAOA approach provides a reasonably high coverage rate, with less overlap and better alteration of the sensor nodes’ configuration, compared to the average coverage rate under the same test conditions.

## 5. Conclusions

This paper suggests an enhanced Archimedes optimization algorithm (EAOA) to solve the wireless sensor network (WSN) nodes’ uneven distribution and low coverage issues in random deployment. Each divided sub-area of the monitoring area of the entire WSN was subjected to node coverage optimization based on the EAOA. The objective function of the optimal node coverage was modeled mathematically by calculating the distance between nodes by measuring each sensor node’s sensing radius and its communication capability in the deployed WSN. The optimization results of multiple sub-areas were fused, combining the sub-areas’ coverage with the complete network node coverage via a mapping mechanism. The updated equations of the EAOA were modified with reverse learning and multidirection strategies to avoid the original drawbacks of the AOA, e.g., slow convergence speed and ease of falling into local extrema whenever dealing with complicated situations. The compared results of the optimal findings on the selected benchmark functions and the WSN node coverage show that the proposed EAOA makes the optimal solution effective for both coverage and benchmark problems. The suggested algorithm will be applied in future works to address WSN node localization [49,50] and optimal WSN deployment [51,52].

## Figures and Tables

**Figure 1 entropy-24-01018-f001:**
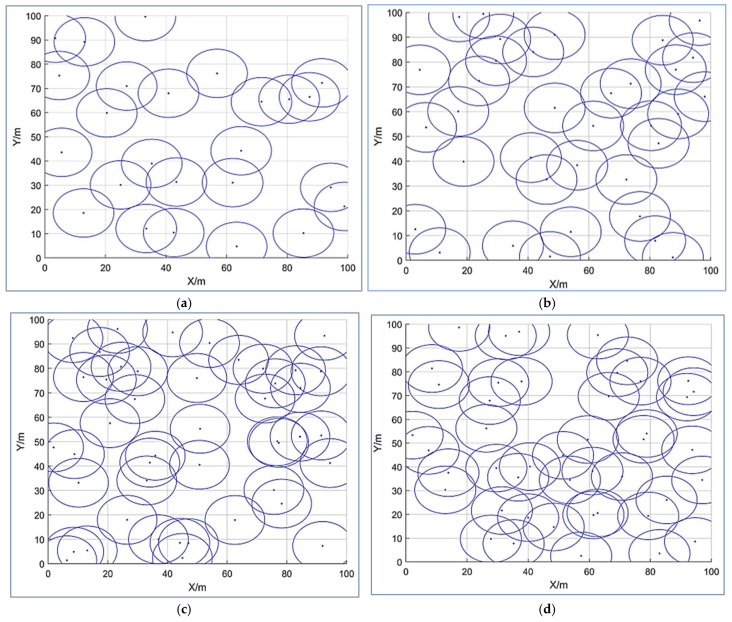
The graphical initialization of the EAOA with the statistical node coverage optimization scheme for different numbers of sensor nodes: (**a**) 20, (**b**) 40, (**c**) 50, and (**d**) 60.

**Figure 2 entropy-24-01018-f002:**
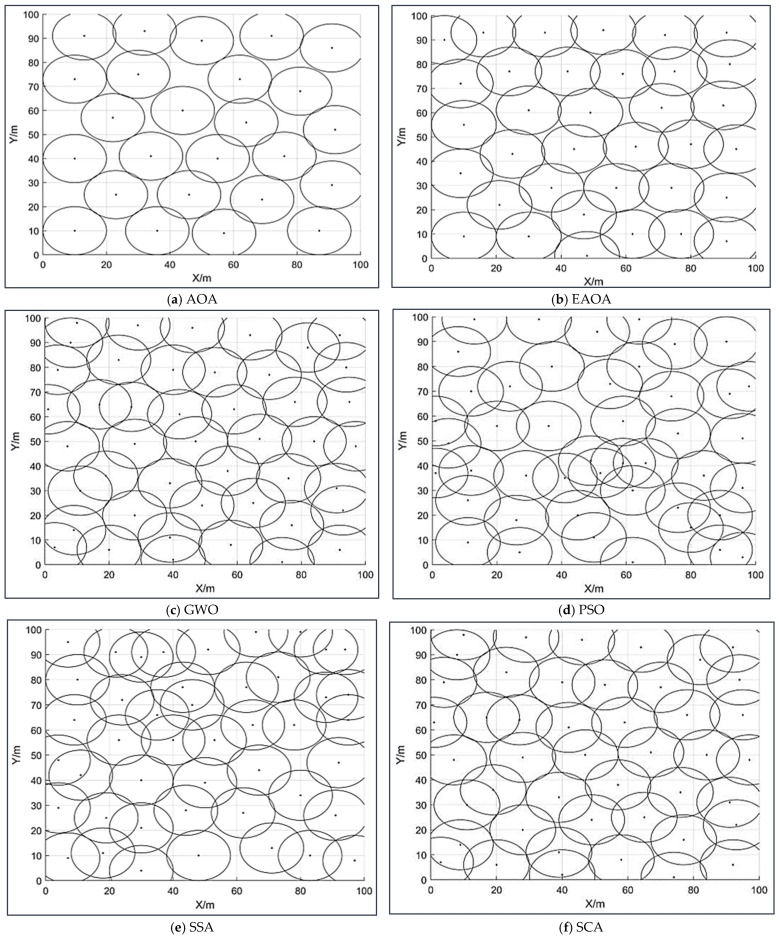
The graphical coverage of six different metaheuristic algorithms for the WSN node area deployment. (**a**) AOA, (**b**) EAOA, (**c**) GWO, (**d**) PSO, (**e**) SSA, (**f**) SCA algorithms.

**Figure 3 entropy-24-01018-f003:**
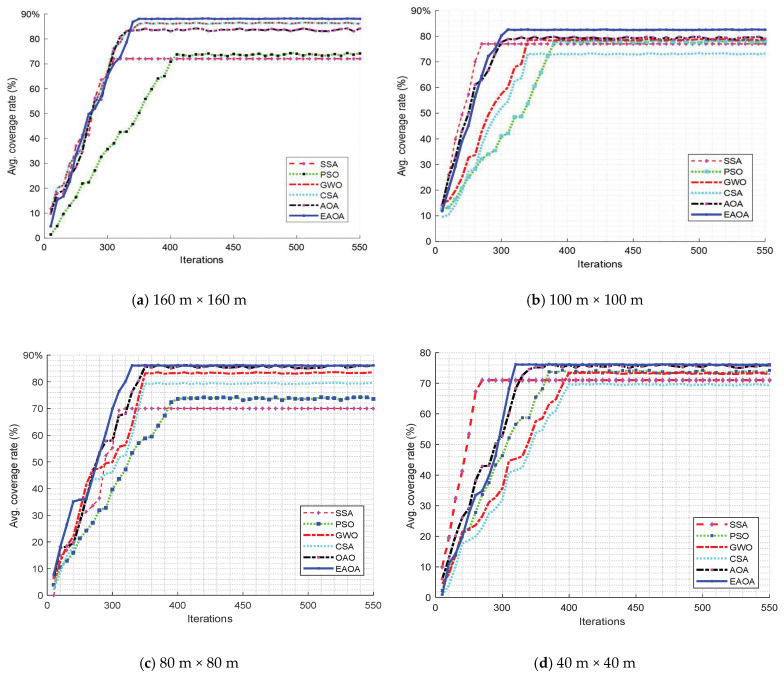
Comparison of the optimal coverage rates of the EAOA with the other schemes in different-sized WSN monitoring node area deployment scenarios. (**a**) 160 m × 160 m, (**b**) 100 m × 100, (**c**) 80 m × 80 m, and (**d**) 40 m × 40 m.

**Figure 4 entropy-24-01018-f004:**
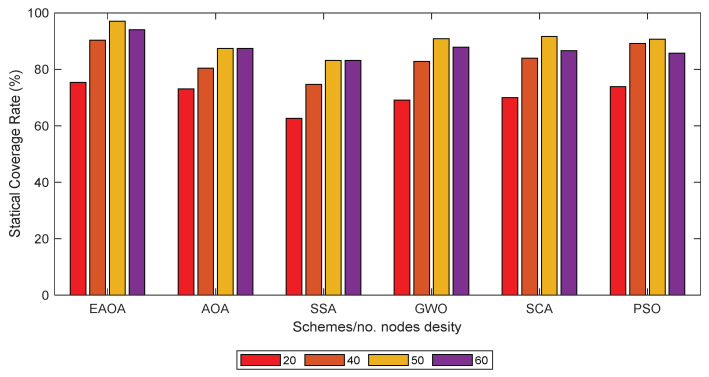
Comparison of the EAOA optimization coverage rates for various sensor node counts deployed in the 2D monitoring of a 100 m × 100 m area.

**Table 1 entropy-24-01018-t001:** Algorithm settings for parameters and variables.

Algorithms	Setting Parameters
EAOA	$C_{1}=2.1,\;C_{2}=5.6,\;C_{3}=1.95,\;C_{4}=0.65$
AOA [41]	$C_{1}=2.1,\;C_{2}=5.6,\;C_{3}=1.95,\;C_{4}=0.65$
GA [25]	$R_{mu}=0.1,\;R_{cr}=0.9$
SA [24]	$P=0.6,\;\alpha =0.8,\;\tau =0.05,\;S_{N}=14.41$
PSO [27]	$V_{max}=10,\;V_{min}=-10,\;\omega =0.9\;to\;0.4,\;c_{1}=c_{2}=1.49455$
PPSO [29]	$G=2,\;R=10,\;V_{max}=10,\;V_{min}=-10,\;\omega =0.9\;to\;0.4,\;c_{1}=c_{2}=1.49465$
PBA [33]	$G=2,\;R=10,\;A_{0}=0.7,\;r_{0}$ =0.15, α=0.25, γ=0.16
FPA [37]	$P_{switch}=0.65,\;\lambda =1.5,\;s_{0}=0.1$
MFO [34]	a=−1, b=1
IMFO [35]	a=−1, b=1 , ω=0.9 to 0.4
WOA [36]	a=2 to 0, b=1, l=[−1,1]
SCA [39]	$r_{1},\;r_{3},\;=rand\left(0,2\right),\;r_{2}\;\in \left[0,\;2\pi \right],\;r_{4}=rand\left(0,1\right)$
ESCA [40]	$r_{1},\;r_{3},\;=rand\left(0,2\right),\;r_{2}\;\in \left[0,\;2\pi \right],\;r_{4}=rand\left(0,1\right)$ , ω=0.9 to 0.5

**Table 2 entropy-24-01018-t002:** The parameter settings for the desired WSN node deployment areas.

Description	Parameters	Values
Desired deployment areas	*W* × *L*	40 m × 40 m, 80 m × 80 m, 100 m × 100 m, 160 m × 160 m
Sensing radius	*Rs*	15 m
Communication radius	*Rc*	20 m
Number of sensor nodes	*M*	20, 40, 50, 60
Number of iterations	*Iter*	500, 1000, 1500

**Table 3 entropy-24-01018-t003:** Comparison of the proposed EAOA method with the other techniques used—i.e., the SAA, PSO, GWO, SCA, and AOA algorithms—in terms of percentage coverage rate, running time, iterations to convergence, and monitoring area size.

Approach	Factor Variables	40 m × 40 m	80 m × 80 m	100 m × 100 m	160 m × 160 m
SSA	Coverage rate (%)	78%	74%	77%	74%
	Consumed execution time (s)	3.09	6.91	7.38	9.34
	No. of iterations to convergence	145	256	234	844
	WSN node numbers	20	40	50	60
PSO	Coverage rate (%)	79%	77%	79%	76%
	Consumed execution time (s)	2.78	6.22	6.65	8.41
	No. of iterations to convergence	396	343	578	754
	WSN node numbers	20	40	50	60
GWO	Coverage rate (%)	80%	80%	84%	78%
	Consumed execution time (s)	3.06	6.84	7.31	9.25
	No. of iterations to convergence	334	44	544	755
	WSN node numbers	20	40	50	60
CSA	Coverage rate (%)	78%	79%	82%	78%
	Consumed execution time (s)	2.92	6.29	7.23	9.22
	No. of iterations to convergence	445	555	665	876
	No. of mobile nodes	20	40	50	60
AOA	Coverage rate (%)	80%	79%	80%	79%
	Consumed execution time (s)	3.12	6.98	7.46	9.44
	No. of iterations to convergence	665	333	563	954
	WSN node numbers	20	40	50	60
EAOA	Coverage rate (%)	80%	82%	87%	80%
	Consumed execution time (s)	2.75	6.15	6.57	8.31
	No. of iterations to convergence	135	503	556	765
	WSN node numbers	20	40	50	60

## Data Availability

Not applicable.

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
