# Peer review of "An Optimal WSN Node Coverage Based on Enhanced Archimedes Optimization Algorithm"

_entropy, 2022, doi:10.3390/e24081018_

Round 1
Reviewer 1 Report
Authors present An Optimal WSN Node Coverage based on Enhanced Archimedes Optimization Algorithm which serves to make an optimal distribution of IoT sensors in 2D space. Authors define IoT sensor as an monitoring device that monitors circle area. For me, it was little bit confusing, because I expected, that the paper concerns with coverage by the signal of LPWAN network. However, the paper is quite well-written and first sections are clear.
The manuscript has following major issues:
1. Results are not written clearly. Please, move tables and figures to appendix and show and describe overall results clearly. Find some interesting points and describe it.
2. Tables are not described clearly. For instance, Avg. is not mentioned in text. Average of what?
3. Add future work to conclusion section.
4. Please, clarify to me and consider manuscript revision in the following questions:
a. You defined coverage of the IoT sensor as a circle and each has coverage with the same radius. The optimal solution should be distribution in grid with same distances. Is it so? Do we need metaheuristic for this task?
b. In results section, you shows the graphs of IoT sensors distribution. Is the global optimum reached? The algorithms do not reach similar results. Why? I think, that for example PSO and GWO can reach very similar results.
The manuscript has also following minor issues:
5. Equations have typographic issues. Please, fix it. You can find typographical rules for scientific texts for example here https://desktop-publishing.web.cern.ch/dtprules.htm
6. There are no units in figures and tables. For example Figure 1, Table 2. Please, fix it.
7. Please use vector graphics. Please check journal requirements for figure font sizes.
8. Reconsider the usage of Microsoft excel for representation of your results (graphs). Using of Microsoft excel will not make a professional impression.
9. Check typography of units. Spaces missing. Check also citation typography, in some occurrences spaces missing.
Author Response
The authors want to thank sincerely Reviewer#1 for the constructive suggestions.
The point-by-point response to the comments is revised and uploaded manuscript.
In the updated manuscript are highlighted fonts indicating changes.

Reviewer 2 Report
- Some parts of the paper need attention regarding the language usage. More specifically, the abstract is not well-written, e.g., “The node coverage optimization also has faced…” must be” The node coverage optimization has also faced”, “…by modifying updating equations …” does not make sense, etc. In the Intro, “Sensor nodes are typically put randomly” can be “Sensor nodes are randomly placed”. A complete language polishing in the whole manuscript must take place.
- The Related Work section contains many mathematical models that are required for the proposed technique. It is more like a section that gives an insight into the system being used by the manuscript’s authors. If this is the case, I would recommend a section “System Definition” which will contain these parts, along with all info regarding the system in use. Then, having described the system the technique is applied on, the Authors can easily suggest their enhanced version in the next section. As for the past works part of the paper it may contain past works given in Introduction and a to-the-point description of the contribution against the current works.
- Results must be better presented. The huge tables are not easy for the reader to understand. These tables must be removed and the results must be given in a more readable fashion (e.g., such as in Figure 1).
- Figure 1 sub-captions cannot be f1, f2, g3, etc. Please use a, b, c, etc.
- Figure 2, Figure 3, and Figure 5 are of low quality in terms of image resolution.
- A high plagiarism rate of 59% has been detected. Many parts are copied by paper in https://ieeexplore.ieee.org/document/9804492
Author Response
We are thankful sincerely to Reviewer#2 for the constructive comments
The point-by-point response to the comments is revised and uploaded manuscript.
In the updated manuscript are highlighted fonts indicating changes.

Round 2
Reviewer 1 Report
Authors answered to my questions, however several questions still remains.
The manuscript still has following minor issues:
1. Equations have typographic issues. Please, fix it. You can find typographical rules for scientific texts for example here https://desktop-publishing.web.cern.ch/dtprules.htm
a. Authors claims, that it is corrected, but not. Please, take a look to for example IEEE transaction journals and find out the math style. rand, max, otherwise etc. must be in roman style. Why you are using vector multiplication instead scalar multiplication? Please fix it.
2. Please use vector graphics. Please check journal requirements for figure font sizes.
a. Some pictures remains in low resolution in bitmap.
3. Reconsider the usage of Microsoft excel for representation of your results (graphs). Using of Microsoft excel will not make a professional impression.
a. Authors says:
i. Author response: Yes, we revised according to your suggestion.
Author action: We updated the manuscript by revising the
b. I appreciate that you accepted the improvement, but you did not any changes. Author action is an incomplete sentence.
4. Check typography of units. Spaces missing. Check also citation typography, in some occurrences spaces missing.
a. Still remains. Please double check.
Author Response
We want to express our gratitude to Reviewer #1 for your valuable and professional comments on our manuscript.
The point-by-point response to the comments is revised and uploaded manuscript. In the updated manuscript are highlighted fonts indicating changes.

Reviewer 2 Report
- Comments have been nicely addressed.
Author Response
We want to thank Reviewer #2 for your valuable and professional comments on our manuscript.
